# Unique and Specific m^6^A RNA Methylation in Mouse Embryonic and Postnatal Cerebral Cortices

**DOI:** 10.3390/genes11101139

**Published:** 2020-09-27

**Authors:** Longbin Zhang, Kunzhao Du, Jing Wang, Yanzhen Nie, Trevor Lee, Tao Sun

**Affiliations:** 1Center for Precision Medicine, School of Medicine and School of Biomedical Sciences, Huaqiao University, Xiamen 361021, China; longbinzhang@fzu.edu.cn (L.Z.); 15659706031@163.com (K.D.); ytwj1994@163.com (J.W.); 2School of Life Sciences and Technology, Shanghai Jiao Tong University, Shanghai 200240, China; yanzhenn@sjtu.edu.cn; 3Department of Cell and Developmental Biology, Weill Medical College, Cornell University, New York, NY 10065, USA; jinyumantang2018@hotmail.com

**Keywords:** m^6^A, RNA methylation, cerebral cortex, neural development, neurological disorders

## Abstract

N^6^-methyladenosine (m^6^A)-mediated epitranscriptomic regulation is critical for various physiological processes. Genetic studies demonstrate that proper m^6^A-methylation is required for mouse brain development and function. Revealing landscapes of m^6^A-methylation in the cerebral cortex at different developmental stages will help to understand the biological meaning of epitranscriptomic regulation. Here, we depict the temporal-specific m^6^A-methylation status in mouse embryonic and postnatal cortices using methylated RNA immunoprecipitation (MeRIP) sequencing. We identified unique m^6^A binding motifs in stage-specific RNAs and found that more RNA transcripts are temporally methylated in embryonic cortices than in postnatal ones. Moreover, we found that cortical transcription factors and genes associated with neurological disorders are broadly as well specifically methylated at m^6^A sites. Our study highlights the importance of epitranscriptomic regulation in the developing cortex and provides a fundamental reference for future mechanistic examinations of m^6^A methylation-mediated gene expression regulation in normal brain development and neurological disorders.

## 1. Introduction

Proper development of the mammalian cerebral cortex is crucial for its morphogenesis and function. In the mouse embryonic cortex, neural stem cells and neural progenitors proliferate and differentiate properly to set up the architectural foundation of the cortical size and shape [1,2]. In the postnatal cortex, migratory neurons form a multiple-layer structure and develop interaction networks within the cortex and with other regions in the central nervous system [3,4]. Temporal and spatial regulation of gene expression precisely drives developing progresses of the cerebral cortex throughout the embryonic and postnatal stages. Altered gene expression causes malformation in the cortex, and results in various neurological disorders, such as microcephaly, macrocephaly and autism [5,6,7].

Accumulating studies have demonstrated essential roles of genetic and epigenetic factors in controlling proper cortical development [8,9]. For instance, aberrance of DNA methylation and/or demethylation has been shown to alter transcription programs of neurogenic genes and lead to brain dysfunctions [10,11,12]. Notably, epitranscriptomic regulations in transcribed RNAs, in particular N^6^-methyladenosine (m^6^A) modification, have been reported to regulate RNA splicing, stability, translation rate and localization [13,14]. In particular, precise regulation of m^6^A-methylation is required for normal cortical development. Genetic deletion of m^6^A methyltransferase Mettl14 results in abnormal proliferation and differentiation of neural progenitor cells in the embryonic cortex [15,16,17]. Moreover, m^6^A-methylation is critical for maturation and function of postnatal and adult brains, as m^6^A-methylation has been shown to regulate dopamine transmission, sensory experience, learning and response to brain injury [18,19,20]. Altered m^6^A-methylation is also associated with neurological disorders, such as Alzheimer’s disease, glioblastoma and major depression [21,22,23,24]. While molecular mechanisms of m^6^A regulation are still unclear, understanding comprehensive profiles of RNA transcripts methylated by m^6^A will facilitate uncovering RNA methylation-mediated gene expression regulation and function.

In this study, to reveal m^6^A-methylation status in the developing cerebral cortex, we conduct methylated RNA immunoprecipitation (MeRIP) sequencing of RNAs collected from mouse embryonic and postnatal cortices. We identify unique m^6^A binding sites in temporally expressed RNAs and reveal specific m^6^A patterns carried by RNAs encoding transcription factors and disease-risk genes in the nervous system. Our results have provided strong evidence and useful reference of the importance of m^6^A-methylation in cortical development.

## 2. Materials and Methods

### 2.1. Animal Care

Wild-type C57BL/6 mice were purchased from Wushi company (Fujian, China). All animal experiments and euthanasia were approved (A2019009) and performed in accordance with the guidelines of Animal Care and Use, Committee of School of Medicine, Huaqiao University.

### 2.2. RNA Extraction and m^6^A-Methylated RNA Immunoprecipitation (MeRIP)

Mice were sacrificed by cervical dislocation and their cerebral cortex was removed and immediately digested with RNAiso^TM^ Plus Reagent (TaKaRa, Dalian, China). Total RNA was extracted using the same reagent according to the manufacturer’s instructions. In general, embryonic-stage RNA was extracted from three E12.5–13 mouse embryos from one litter, and postnatal-stage RNA was extracted from three P14 mice from one litter. About 300 μg RNA was used for MeRIP, and all RNA samples were treated with DNase to remove genomic DNA.

The m^6^A-MeRIP reaction was performed according to a previous description [25]. Briefly, total RNA was chemically fragmented to 100–200 nucleotides (for m^6^A-seq). A portion of fragmented RNA was saved as input before immunoprecipitation (IP). The remaining fragmented RNA was incubated with an m^6^A-specific antibody (Synaptic Systems, cat. No. 202 003) at 4 °C for 2 h, followed by addition of Immobilized Recombinant Protein A (Repligen, cat. No. IPA300) for immunoprecipitation. After 2-h incubation at 4 °C, beads were extensively washed three times with IP Buffer (150 mM NaCl, 0.1% Igepal CA-630 and 10 mM Tris-HCl, pH 7.4). Immunoprecipitated RNA was recovered through elution with m^6^A nucleotide followed by ethanol precipitation. Due to the recognition feature of the m^6^A antibody, the detection of methylated sites might contain both m^6^A and m^6^A(m) [26]. Additionally, IgG (Sigma-Aldrich, cat. No. I8765) was used as a negative control of the immunoprecipitated reaction.

### 2.3. m^6^A Sequencing (m^6^A-seq) and Data Analysis

For high-throughput sequencing, purified RNA fragments from m^6^A-MeRIP were used for library construction with the NEBNext Ultra RNA library Prep kit from Illumina and were sequenced with Illumina HiSeq 4000. Library preparation and high-throughput sequencing were performed by Novogene (Beijing, China). The m^6^A-seq data (NCBI GEO: GSE141938) was analyzed with reference to the previous standardized pipeline [27]. Briefly, the m^6^A-seq data was cleaned by trimming the adapter using trim galore. The cleaned reads were aligned to the house mouse (*Mus musculus*) genome UCSC mm10 using Bowtie2 with default parameters and were then converted to a bam file by samtools. The PCR duplicates in the aligned data were removed by the MarkDuplicates function of Picard (version 1.95) with parameters, REMOVE_DUPLICATES = Ture. After filtering the duplicates, samtools was used to sort the bam file according to the coordinates. The m^6^A peaks were detected by MACS (Version 1.4.0rc 2) with parameters, -format = “BAM” -gsize = 282000000 -tsize = 101 -nomodel -shiftsize =150 -to-large False -w -S. The resulting m^6^A-seq data from m^6^A-antibody immunoprecipitation samples at E12.5–13 and P14 were normalized to their respective input RNA data before being used for the following analysis.

Hypergeometric Optimization of Motif EnRichment (HOMER) was used for m^6^A motif analysis. Briefly, we set the peak-harboring genome interval as detected data, while the peak-removed exon interval was set as background data. The detected data and background data were compared for motif analysis using HOMER with the following orders: -gc, -RNA, -len 6, -size given, -homer2, -p 2, -S 25, -dumpFasta. Each sample was independently measured, followed by intra-group comparison in E12.5–13 triplicates and P14 triplicates, respectively.

The integrative genomics viewer (IGV) tool was used for visualization of m^6^A peaks along the whole transcript [28]. Notably, we identified a methylation site as significant if the mean IGV value of peak was more than 3, which is approximately equal to log10 (500) (500 was the threshold of peak scores). Even if the IGV presentation showed a visible peak, the site was considered as non-methylated if the IGV value was less than 3. The heatmaps of the fold enrichment score for m^6^A peaks, the Volcano maps of transcript levels and the Venn analysis of targeted genes were all performed using the Omic Share tools, a free online platform for data analysis (http://www.omicshare.com/tools).

The DAVID tool [29] was used for the Gene ontology (GO) and the Kyoto Encyclopedia of Genes and Genomes (KEGG) analysis by applying default parameters. Color intensity indicated the value of −log_10_ (*p* value), while the length of columns or the size of the circles indicated the gene counts. A full list of all selected terms of the biological process, cellular components, molecular functions and KEGG pathway category are provided in the Appendix A.

### 2.4. Statistics

All data are shown as mean ± SEMT. One-way analysis of variance (ANOVA) with post hoc contrasts were used for statistical analysis. The results were considered significant at probability level less than 0.05.

## 3. Results

### 3.1. m^6^A-Methylation Profiles in Embryonic and Postnatal Cerebral Cortices

Cortical development is a continuous process of neural progenitor proliferation, differentiation and newborn neuron migration and maturation [1,3,30]. If m^6^A-methylation is essential for cortical development, we suspected that m^6^A-methylation profiles should reflect the dynamic process in developing cortices. Because mouse cortices at embryonic days 12.5 to 13 (E12.5–E13), termed here as embryonic stage (E-stage), and cortices at postnatal day 14 (P14), termed as postnatal stage (P-stage), consisted of more neural progenitors and mature neurons, respectively, we performed MeRIP sequencing (MeRIP-Seq) using cortical tissues from E12.5–E13 and P14 mouse brains (Figure 1A).

We identified an average of 27,113,335 and 25,574,954 clean reads from E-stage and P-stage cortices, respectively. The coverage was 545 and 521 reads/1000 bp in E-stage and P-stage cortices, respectively [26,31]. Pearson correlation analysis showed related a co-efficiency approximately to 1 among each group of three replicates at each stage, indicating that MeRIP-Seq data displays high consistency and low variation (Appendix A). Each read was frequently mapped to transcribed RNAs, and was clustered as a distinct peak, which was converged to about 100 nucleotides (nts) wide fragment with a summit in the center (Appendix A). After normalization to the input data, we detected 8355, 8488 and 8451 methylated RNAs from three E-stage cortices, and 8365, 8373, and 8151 methylated RNAs from three P-stage cortices (Appendix A). We identified 4733 and 4449 intersecting m^6^A-methylated RNAs with high-confidence in all three E-stage and P-stage replicates, respectively (500 value of sequencing peak-scores and 0.001 *p*-value were set as threshold limit) (Appendix A, Appendix A).

To identify m^6^A motif in temporal-specific cortical RNAs, we applied a sensitive and unbiased tool, Hypergeometric Optimization of Motif EnRichment (HOMER), for motif discovering. We detected the most confident motif, GGGACA, which was highly enriched in m^6^A peaks from triplicated E-stage cortices (Figure 1B). Motif variants, such as GGACUG and UGACU[GC], which shared a conserved sequence, GAC, with m^6^A in the middle, were also predicted. Similarly, we also detected the most confident motif, GGACUG, and predicted variants, UAGGAC and GACU[GA][AG], in m^6^A peaks from triplicated P-stage cortices (Figure 1C). These results indicate that m^6^A-methylation displays unique motifs in RNAs from embryonic and postnatal cortices.

### 3.2. None or Weak Correlations between m^6^A-Methylation Profile and Transcription Profile

We analyzed RNA sequencing data of E- and P-stage cortices (NCBI GEO: GSE116056), and identified 2970 RNA transcripts from E-stage neural progenitors displaying higher expression levels than those from P-stage cortices, while 5416 transcripts showed higher expression from P-stage cortices than those from E-stage ones (Figure 2A). We detected 32% of 2970 RNAs and 25% of 5416 RNAs containing m^6^A-methylation sites in E-stage and P-stage cortices, respectively (Figure 2B,C). These data suggest that only parts of temporally-highly expressed RNAs are methylated at the m^6^A site. In addition, correlation analysis between levels of RNA transcription and m^6^A-methylation showed mixed distribution, suggesting none or weak correlations between two aspects (Figure 2D,E).

To comprehensively evaluate the correlation between the transcribed change of RNAs and the position of m^6^A, the amounts of m^6^A sites were calculated according to their distribution region in RNAs with different changes in their transcription level (Appendix A). We found that the up-regulated RNAs in postnatal stage harbor more postnatal-special m^6^A sites than embryonic-special m^6^A sites in RNA regions of 5′ untranslated region (5′UTR), coding sequence (CDS) and 3′UTR (Appendix A). On the contrary, both down-regulated and none-changed RNAs had more embryonic-special m^6^A sites than postnatal-special m^6^A sites in targeted regions (Appendix A). Additionally, the amounts of common m^6^A sites (detected in both embryonic and postnatal stage) were generally higher than of E-special and P-special m^6^A sites in targeted regions of both changed and none-changed RNAs (Appendix A). Our analysis indicates that the transcription changes of RNAs show partial correlation with temporal-specificity of m^6^A rather than the position of m^6^A.

### 3.3. Temporally Methylated RNAs

We next analyzed distributions of m^6^A summits among transcribed RNAs residing in different chromosomes. We observed that there are, overall, slightly higher m^6^A summits in RNAs from E-stage cortices than those from P-stage, suggesting a higher global methylation level in embryonic cortices (Appendix A). Moreover, we counted numbers of m^6^A-methylated RNAs in each chromosome and found that more m^6^A-methylated RNAs were detected in E-stage cortices than in P-stage ones in most chromosomes (Appendix A). These results further suggest that m^6^A-methylation appears to be more active in embryonic cortices than in postnatal ones.

We further compared RNAs that were temporally and continuously methylated either in E-stage cortices or in P-stage ones. We identified 26% embryonic stage-specifically methylated RNAs (SMRs), termed as E-SMRs, 18% postnatal stage specifically methylated RNAs, named P-SMRs, and 56% continuously methylated RNAs, termed as CMRs, at both stages (Figure 1D). Notably, there were more E-SMRs than P-SMRs (about 1.5 times). These results suggest that the majority of RNAs are continuously methylated in the developing cortex, and more RNAs are temporally methylated in embryonic cortices than in postnatal ones.

To exclude an effect of input signals, we further identified significantly methylated RNAs by analyzing the fold enrichment (FE) score, which is calculated as peak-reads of immunoprecipitation samples versus those of relative input samples. We detected 3061 CMRs, 605 E-SMRs and 531 P-SMRs (*p* < 0.001, peak score > 500, FEs > 2.0) (Figure 1E). Moreover, similar to all methylated RNAs, the number of E-SMRs was more than that of P-SMRs, and E-SMRs showed significant enrichment in chromosomes 7, 11 and 17 (Figure 1F). These data suggest that many RNAs from different chromosomes are temporally modified by m^6^A-methylation in the developing cortices.

### 3.4. Patterns of m^6^A-Methylation within Transcribed RNAs

To examine the distribution pattern of m^6^A binding sites within transcribed RNAs, we first analyzed the number of m^6^A binding sites in them. In E-stage cortices, one single m^6^A peak was detected in 58% RNAs, two m^6^A peaks in 26% RNAs, three m^6^A peaks in 10% RNAs, and four or more m^6^A peaks in 6% RNAs (Appendix A). In addition, one single, two, three and four or more m^6^A peaks were observed in 61%, 25%, 9% and 5% RNAs transcribed in P-stage cortices, respectively (Appendix A). These results indicate that a majority of cortical RNAs contains one or two m^6^A binding sites.

We then determined the distribution position of m^6^A binding sites in RNAs. We observed that m^6^A binding sites resided in coding sequences (CDS) in 47.02% and 45.85% RNAs from E-stage and P-stage cortices, respectively, and in untranslated regions (UTRs) in 36.62% and 37.12% RNAs from E-stage and P-stage cortices, respectively (Figure 3A). Interestingly, m^6^A binding sites were more abundant in the 3′UTR than in the 5′UTR in both stages (Figure 3A). Moreover, m^6^A binding sites were also detected in regions of the stop codon in many RNAs. To further investigate the distribution patterns of m^6^A mapping to genomic region, we calculated the numbers of m^6^A peaks mapped to specific genomic regions versus numbers of all detected m^6^A peaks (Figure 3B). Significant differences of m^6^A distributions mapping to genomic regions from E-stage and P-stage cortices were mostly located in the upstream region (5–10 kb), downstream of CDS (2 kb), near stop codon (NSC, 200–500 bp), 3′UTR and transcriptional termination site (TTS, 500–1000 bp) (Figure 3B). Additionally, we also calculated the percentage of m^6^A summits in different regions of mRNAs versus the numbers of all detected summits and found an enrichment of m^6^A summits in the region of near stop codon (NSC) and 3′UTR regions (Appendix A). Collectively, these data indicate variations of m^6^A-methylated sites in transcribed RNAs, beside the CDS, with a high enrichment in the NSC and 3′UTR regions.

### 3.5. Functional Association of m^6^A-Methylation with Transcribed RNAs

We found that m^6^A binding sites show enrichment in the NSC and 3′UTR regions, and within these regions, more m^6^A binding sites are detected in RNAs from P-stage than those from E-stage cortices (Figure 3B). Therefore, we focused the remaining analyses on RNAs with enriched m^6^A binding sites in the NSC and 3′UTR regions. We first performed a gene ontology (GO) analysis on E-SMRs and P-SMRs (Figure 4A, Appendix A). The top categories of biological process (BP) for E-SMRs were mainly involved in regulation of transcription, cell differentiation and cell cycle, while they were associated with ion transport, signal transduction and cell adhesion for P-SMRs (Figure 4A). Interestingly, three common categories of nervous system development, cell-cell signaling and oligodendrocyte differentiation were identified in both E-SMRs and P-SMRs (Figure 4A).

Moreover, we analyzed cellular component (CC) and found that enriched categories for E-SMRs are associated with DNA-replication and RNA-synthesis, and those for P-SMRs are enriched in cell membrane construction and assembly of specific neuronal cell bodies, dendrites and synapses (Figure 4B). In addition, molecular function (MF) analyses showed that categories for E-SMRs are generally enriched in DNA and ion binding, and those for P-SMRs are involved in channel activities (Figure 4C). GO analyses for CMRs showed enriched categories, mainly in fundamental biological processes, such as transport, membrane and protein binding (Figure 4D). Collectively, our results indicate that temporal-specific m^6^A-methylation is highly associated with E-SMRs that function in transcription and cell cycle regulations, and with P-SMRs that play a role in regulating differentiation and signal transduction, which is consistent with major biological events occurring in neural progenitors and mature neurons.

### 3.6. Status of m^6^A-Methylation in Genes Involved in Major Signaling Pathways

Because cortical development is precisely modulated by several signaling pathways, we examined whether genes in major pathways are regulated by m^6^A-methylation. We first analyzed E-SMRs with m^6^A binding sites in the NSC and 3′UTR and identified 27 enriched pathways using the Kyoto Encyclopedia of Genes and Genomes (KEGG) analysis (Appendix A). In particular, genes in the PI3K-Akt signaling pathway, hippo signaling pathway, p53 signaling pathway and hedgehog signaling pathway displayed significant m^6^A-methylation (Appendix A).

Among these pathways, hippo and hedgehog signaling pathways have been shown to play crucial roles in cortical development [32,33]. Hence, we investigated m^6^A-methylation status in major genes among E-SMRs in these two pathways. Studies have shown that Par3/Par6/aPKC complex, Lats1, YAP and Tead1 are associated with the hippo-YAP signaling pathway [34,35]. We analyzed their m^6^A-methylation patterns using the Integrative Genomics Viewer (IGV) (Figure 5). We detected one m^6^A binding site near stop codon of Par3 and Par6 transcripts (Figure 5A). Moreover, Lats1 displayed one m^6^A binding site near the stop codon, and Tead1 had one m^6^A site in the 3′UTR. In addition, studies have shown association of the Wnt and hippo-YAP signaling pathways [36]. We found that Wnt5a/b, Wnt8b and Fzd2 are methylated near their stop codons with one or two m^6^A sites. Furthermore, we detected one or more m^6^A sites in the NSC or 3′UTR region in Smo, Sufu, and Gli1/2/3 in the hedgehog signaling pathway (Figure 5B and Appendix A). Collectively, our data indicate that m^6^A-methylation appears to be a common event in genes which are involved in major signaling pathways regulating cortical development.

Furthermore, we examined the m^6^A-methylation status in P-SMRs with m^6^A binding sites in the NSC and 3′UTR using the KEGG analysis. We observed that neural signaling transmissions, such as neuroactive ligand-receptor interaction pathways and multi-chemical synapses, are specifically enriched in P-SMRs (Appendix A and Appendix A). In detail, we detected 20 P-SMRs involved in the glutamatergic synapse pathway, 16 in the cholinergic synapse pathway, 13 in the GABGergic synapse pathway, 12 in the serotonergic synapse pathway, and 10 in the dopaminergic synapse pathway (Appendix A). We also noticed that a majority of P-SMRs involved in the above pathways encode receptors located in the postsynaptic membrane; in particular, P-SMRs involved in the glutamatergic synapse pathway are coordinated with calcium ion (Ca^2+^) and the calcium signaling pathway (Appendix A). In addition, these targeted genes had multiple m^6^A sites along their transcripts (Appendix A), suggesting an important role of m^6^A-methylation in gene functional regulation. These analyses indicate that m^6^A methylated genes in P-stage cortices are mostly involved in signaling transmission of postsynaptic membrane reception.

### 3.7. Cortical-Specific Transcription Factors Methylated at the m^6^A Sites

Because transcription factors play crucial roles in cortical development and function [37], we next focused on analyzing m^6^A-harboring status in RNAs encoding cortical-specific transcription factors with m^6^A binding sites near their stop codon and the 3′UTR.

Among 930 E-stage-specifically methylated RNAs, we detected 60 genes functioning as transcription factors such as PAX6, EMX2 and TBR2. Pax6 and Tbr2 harbored one m^6^A site near the stop codon, and Emx2 contained one m^6^A site near stop codon and one in the 3′UTR, illustrated by the IGV (Figure 6A). Moreover, we detected 17 transcription factors among 794 P-SMRs (Appendix A). For instance, Satb2 and Sox10 harbored one m^6^A site near the stop codon, and Olig1 had two m^6^A sites near the stop codon (Figure 6B). 

In addition, we analyzed 2562 CMRs, identified 71 transcription factors expressed in both E- and P-stage cortices, and found 54 of them harbored one m^6^A site in the NSCs or 3′UTR region (Appendix A). For example, Emx1 and Tbr1 had one m^6^A site, and Cux2 and Neurod2 displayed two m^6^A sites (Figure 6C).

Besides the NSC and 3′UTR region, there were 569 RNAs harboring m^6^A sites at the 5′UTR and 1939 RNAs in the CDS region. Among them, 13 of 194 E-SMRs, 9 of 141 P-SMRs and 10 of 234 CMRs were RNAs encoding transcription factors harboring m^6^A sites in the 5′UTR region, while 51 E-SMRs, 12 P-SMRs and 46 CMRs were transcription factors with m^6^A sites in the CDS region (Appendix A). Similarly, more than one m^6^A site was detected from the targeted transcripts (Appendix A).

Taken together, our data indicate that cortical transcription factors are broadly methylated at the m^6^A site.

### 3.8. Detection of m^6^A-Methylated RNAs Related to Neurological Disorders

Previous studies have identified pathogenic genes associated with neurological disorders [38], so we next examined m^6^A-methylation patterns of these genes. We identified 45 disease-risk genes modified by m^6^A-methylation that is restricted to the NSC or the 3′UTR region. In total, 13 E-SMRs were reported to be associated with neurodevelopmental diseases, such as microcephaly, polymicrogyria and megalencephaly (Appendix A). We observed one single m^6^A site near the stop codon in 7 microcephaly-related E-SMRs, including Brca2, Cep152, Ddx11, Nde1, Kif14, Stil and Cdk5rap2, 3 polymicrogyria-related E-SMRs (Eomes/Tbr2, Pax6 and Foxp2), and 3 megalencephaly-related E-SMRs (Gli3, Ccnd2 and Kif7) (Figure 7A, Appendix A). Moreover, Cdk5rap2 and Foxp2 contained two m^6^A sites near the stop codon and in the 3′UTR region (Figure 7B).

Additionally, we detected 4 P-SMRs involved in Alzheimer’s disease (Psen2), Parkinson’s disease (Dnajc6) and major depression disorder (Htr2a and Htr5a) (Appendix A). Htr5a contained two m^6^A sites located near stop codon and in the 3′UTR, and the remaining 3 P-SMRs harbored one m^6^A site near stop codon (Figure 7C, Appendix A).

Moreover, we identified 28 CMRs associated with neurological disorders (Appendix A). Among them, Mapt, Nlgn3 and Mtor contained two m^6^A sites in the NSC or 3′UTR, Nin displayed three sites, and the rest of the CMRs contained one m^6^A site in the NSC or 3′UTR (Figure 7D, Appendix A). Our analyses indicate that the targeted RNAs associated with neurological diseases are modified by m^6^A-methylation with a variation of m^6^A binding sites.

Furthermore, we found that disease-risk RNA transcripts also display m^6^A-methylation sites in the 5′UTR and CDS. For example, 2 CMRs, Mapt and Foxg1, had one m^6^A-methylation site in the 5′UTR, microcephaly-related genes, Brca1 and Ddx11, had one m^6^A site, and Aspm and Pcnt harbored four or more m^6^A sites within the CDS (Figure 7A, Appendix A). The data suggest that m^6^A-methylation also modulates pathogenic RNAs via the 5′UTR and CDS region.

## 4. Discussion

Development of the cerebral cortex is precisely controlled by both genetic and epigenetic regulations. Epitranscriptomic regulation mediated by m^6^A is a prominent RNA modification and plays a crucial role in various biological processes. In this study, we uncovered the m^6^A-methylation status in mouse embryonic and postnatal cortices using MeRIP sequencing and identified distinct m^6^A binding motifs. While m^6^A-methylation appears to be a common phenomenon in transcribed RNAs associated with transcription regulation and disease conditions, it displays temporal and functional specificity. Our study highlights a fundamental reference of epitranscriptomic regulation in the developing cortex.

Besides the cerebral cortex, m^6^A-methylation occurs in the whole brain, including the cerebellum, suggesting a common epitranscriptomic modification in the central nervous system [39,40,41]. Dynamic m^6^A modification is reflected not only in region-specificity between the cerebellum and cerebral cortex [39], but also in temporal-specificity. Through comparing m^6^A-methylation between embryonic and postnatal cortices, we have found a generally higher methylation status in embryonic cortices than in postnatal cortices, which may reflect a more methylated gene expression regulation in embryonic neural progenitors and differentiated neurons [15]. Moreover, m^6^A-methylation displays distinct motifs in RNAs from embryonic and postnatal cortices. Despite this difference, the core sequence of m^6^A motifs (GGAC) in cortices at each stage is in accordance with previous studies of the mouse cerebellum and cortex [39]. Our finding suggests that temporal methylation of m^6^A is likely achieved by binding to specific sites in transcribed RNAs specifically functioning in neural progenitors and mature neurons. Finally, while m^6^A-methylation occurs in the coding sequence in an RNA, it appears more frequently near the stop codon and in the 3′ UTRs. Our finding is consistent with previous reports of m^6^A binding sites [40,42].

We also found that m^6^A-methylated RNAs display distinct functions in embryonic and postnatal cortices. While genes specifically methylated in embryonic cortices function in cell differentiation, DNA repair and cell cycle, those in postnatal cortices play roles in transport and synapse assembly. Genes methylated at both stages function mostly in chromatin modification, nucleus and membrane structures, which represent fundamental cellular functions. Similarly, m^6^A-modified RNAs in the cerebellum are functionally enriched in cell cycle and DNA repair at early developing stages and changed to signal transduction and synaptic plasticity at later developing stages [41]. These studies and ours reveal that RNA methylation is in parallel with both cortical and cerebellar developmental progresses, with gene expression patterns that control these steps. An interesting question is that whether RNA methylation is a default event that occurs in dynamically expressed genes or an internal drive that participates temporal gene expression regulation. Future work will be to decipher RNA methylation-mediated gene expression phenomenon.

Genetic studies of depleting either m^6^A methyltransferases, such as METTL3 and METTL14 or demethylase FTO (fat mass and obesity associated protein), have demonstrated their important roles in controlling cortical development and regulating brain cognitive functions [16,43,44]. Our study has provided a strong reference supporting the cause of those phenotypes. For example, PAX6 and TBR2 control proliferation of neural progenitors, and TBR1 and SATB2 regulate migration and maturation of postmitotic neurons [45,46,47]. We found that the RNAs encoding these transcription factors are harboring m^6^A sites, even though the position and number of methylation sites are different among them. Moreover, Par3 and Par6 are involved in the hippo-YAP signaling pathway and function in controlling polarity of cortical neural progenitors [48,49]. Our study has demonstrated m^6^A-methylation in these and other key genes in the hippo pathway. Furthermore, we found that major genes associated with neurological disorders also are methylated at the m^6^A site. Even though genetic mutations in these genes are associated with disorders, such as microcephaly-related genes, it is unclear why m^6^A-methylation seems to be a common event in them and how it may modulate proper expression and translation of these disease genes.

## 5. Conclusions

Our study has uncovered an image of m^6^A-methylation in the mouse developing cortex and revealed the basic rules of m^6^A-methylation in different genes highly expressed in the cortex. Our finding has provided an informative reference for future mechanistic examinations of m^6^A methylation-mediated gene expression regulation in normal brain development and neurological disorders.

## Figures and Tables

**Figure 1 genes-11-01139-f001:**
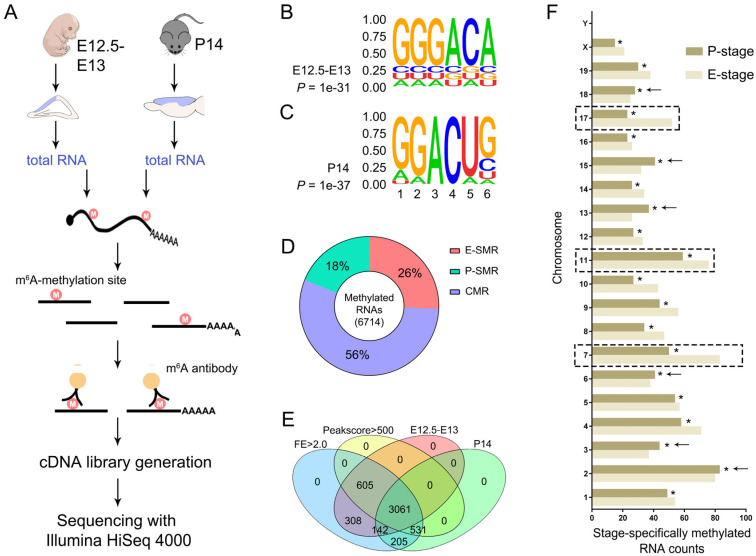
Profiles of m^6^A-methylation in mouse embryonic and postnatal cerebral cortices. (**A**) Schematic presentation of methylated RNA immunoprecipitation (MeRIP) sequencing. Total RNA was extracted from the cerebral cortex (blue) of embryonic day 12.5–13 (E12.5–E13) and postnatal day 14 (P14) mice. (**B**,**C**) Sequence motifs identified within m^6^A peaks. The motif GGGACA and GGACUG were highly enriched in m^6^A peaks from E12.5–E13 and P14 samples, respectively. (**D**) The proportion of embryonic stage-specifically methylated RNAs (SMRs), termed as E-SMRs, postnatal stage-specifically methylated RNAs, named P-SMRs, and continuously methylated RNAs at both stages, termed as CMRs. (**E**) Identification of temporally-specifically methylated RNAs. The parameter value was limited as threshold. Peak score values > 500, fold enrichment > 2.0. *p*-values for all presenting RNAs were less than 0.001. (**F**) The counts of temporally-specifically methylated RNAs in each chromosome. The counting numbers for E-SMRs were significantly higher in chromosomes 7, 11 and 17 (dashed boxes), and for P-SMRs were higher in chromosomes 2, 3, 13, 15 and 18 (arrows). The asterisked bar in each group differ significantly from those unmarked (Tukey’s HSD (Honestly significant difference), *p* < 0.05).

**Figure 2 genes-11-01139-f002:**
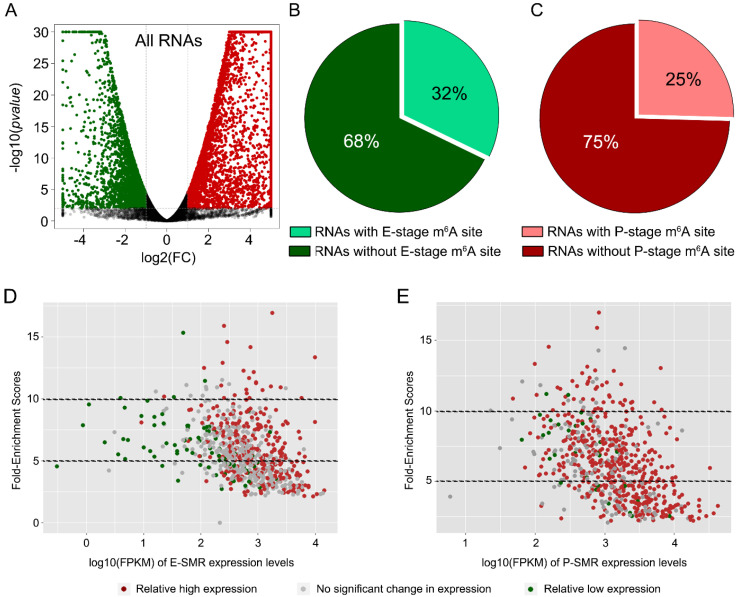
None-correlation between fold enrichment of MeRIP sequencing and reads of transcriptome sequencing. (**A**) The Volcano map of transcription profiles of all RNAs in P-stage compared to E-stage. The transcriptome data were downloaded and used from (NCIB GEO: GSE116056). (**B**) The proportion of down-regulated RNAs with or without an E-stage m^6^A site. (**C**) The proportion of up-regulated RNAs with or without a P-stage m^6^A site. (**D**,**E**) The correlation analysis of embryonic stage-specifically methylated RNAs (E-SMRs) or postnatal stage-specifically methylated RNAs (P-SMRs) between their fold enrichment score values of m^6^A-methylation levels and the FPKM reads of their expression levels.

**Figure 3 genes-11-01139-f003:**
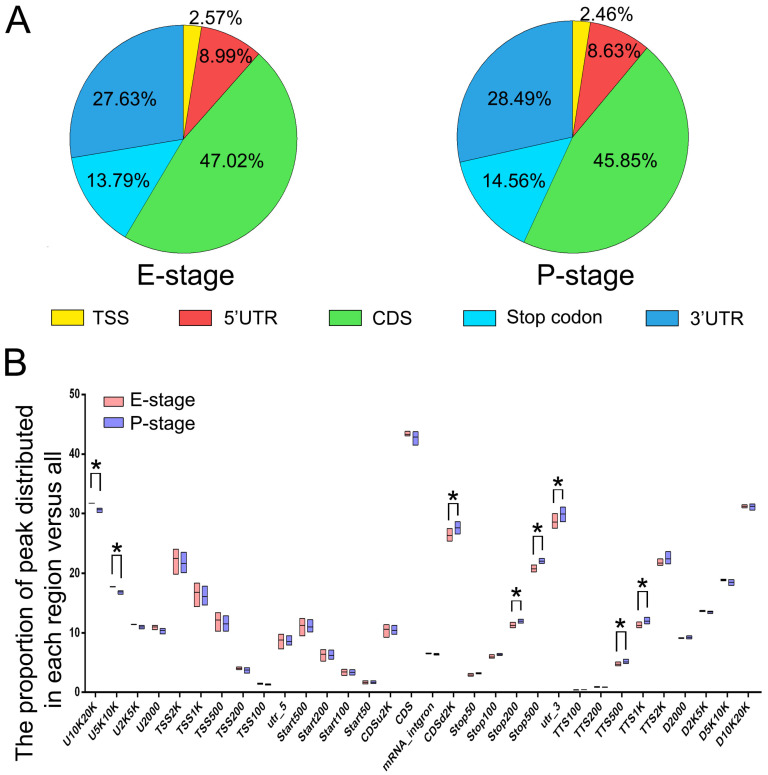
Patterns of m^6^A-methylation within transcribed RNAs. (**A**) The proportion of m^6^A peaks distributed among five cortical RNA regions at E- or P-stage. (**B**) The proportion of m^6^A peaks distributed in the genome. U: upstream; D: downstream; TSS: translational start site; TTS: translational termination site; UTR: untranslated region; start: translation start codon; stop: translation stop codon; CDS: coding sequence; K: kilobase. The U2000 and D2000 refer to 2000 bp upstream of genes and 2000 bp downstream of genes, respectively. CDSu2k and CDSd2k refer to 2KB upstream and downstream of CDS, respectively. Tss100, TTS100, Stop100 and Start100 refer to the range of 100bp centered on TSS, TTS, start codon and stop codon. There might be overlapping parts among some of the regions above. The asterisked bar in each group differ significantly from those unmarked (Tukey’s HSD, *p* < 0.05).

**Figure 4 genes-11-01139-f004:**
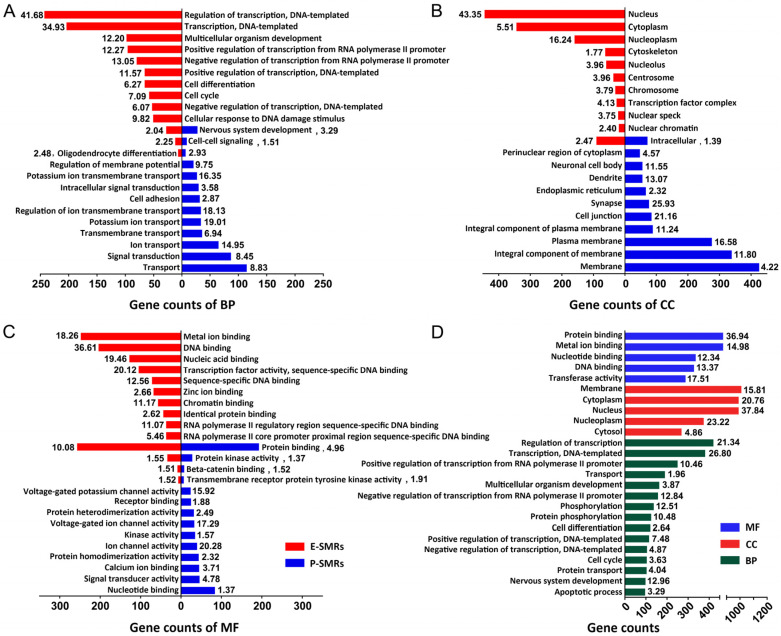
Functional association of m^6^A-methylation with transcribed RNAs. (**A**–**C**) Gene ontology (GO) analysis on E-SMRs and P-SMRs in categories of biological process (BP), cellular component (CC) and molecular function (MF). (**D**) GO analysis on continuously methylated RNAs (CMRs). The –log10(*p*-Value) of each category is shown by the respective box.

**Figure 5 genes-11-01139-f005:**
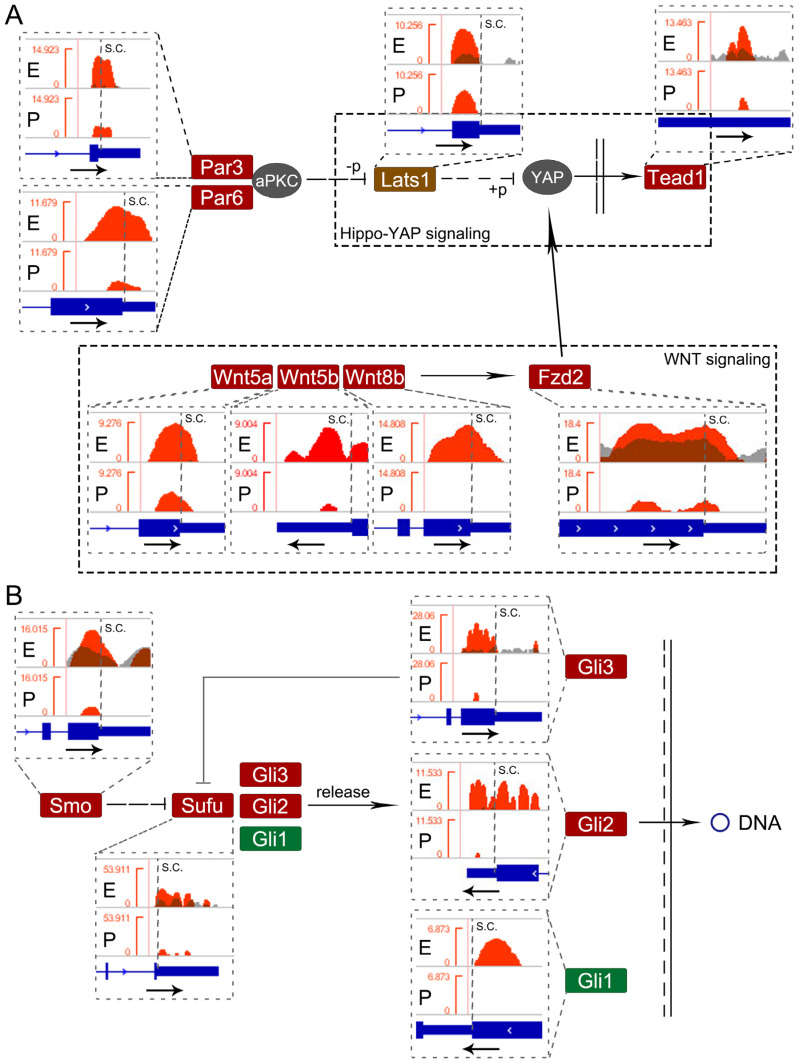
Status of m^6^A-methylation in genes in the hippo-YAP pathway and hedgehog pathway. (**A**) Integrative Genomics Viewer (IGV) analysis of four E-SMRs (Par3, Par6, Lats1 and Tead1) in the hippo-YAP pathway. Four Wnt-related E-SMRs (Wnt5a/b, Wnt8b and Fzd2) participated in the hippo-YAP pathway. (**B**) IGV analysis of five E-SMRs (Smo, Sufu, Gli1/2/3) in the hedgehog pathway.

**Figure 6 genes-11-01139-f006:**
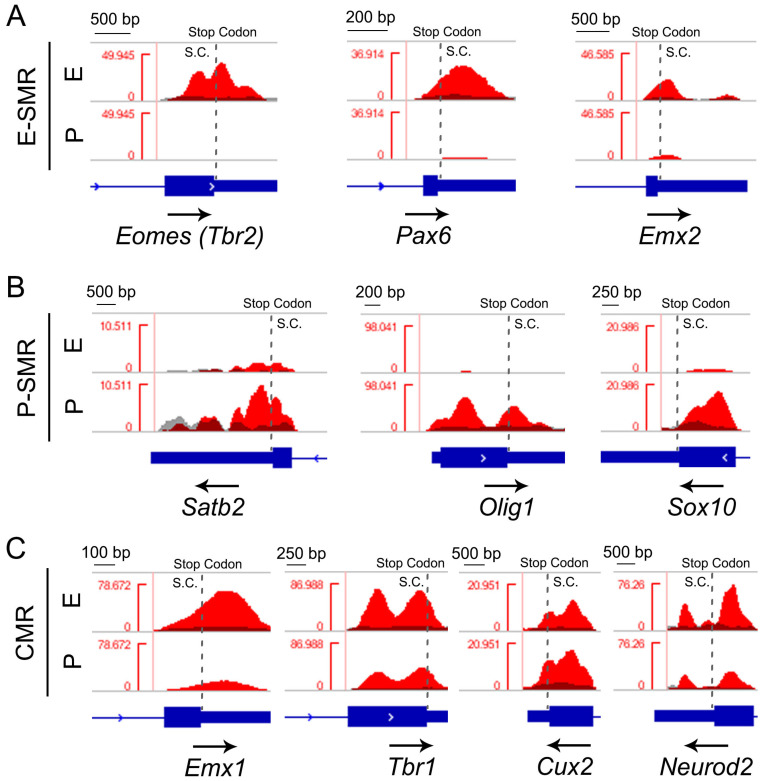
Status of m^6^A-methylation in cortical transcription factors. (**A**) Integrative Genomics Viewer (IGV) analysis of three E-SMRs that function as transcription factors (Pax6, Emx2 and Tbr2). (**B**) IGV analysis of three P-SMRs (Satb2 Olig1 and Sox10). (**C**) IGV analysis of four CMRs encoding transcription factors.

**Figure 7 genes-11-01139-f007:**
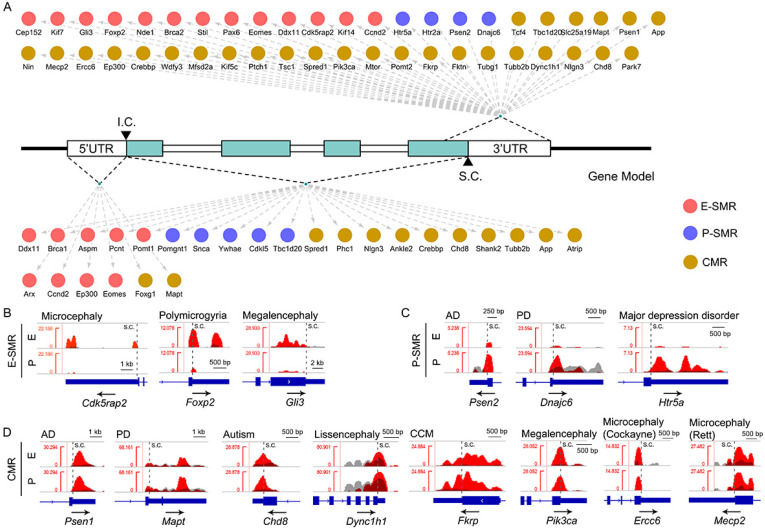
Status of m^6^A-methylation sites of genes in neurological disorders. (**A**) Identification of methylated genes and their specifically methylated patterns in RNA transcripts. The red dots stand for E-SMRs, the blue dots stand for P-SMRs, and the dark yellow dots stand for CMRs. (**B**) IGV analysis of three representative E-SMRs that act as pathogenetic genes (Cdk5rap2, Foxp2 and Gli3) associated with neurological disorders. (**C**) IGV analysis of three representative P-SMRs (Psen2, Dnajc6 and Htr5a). (**D**) IGV analysis of eight representative CMRs (Psen1, Mapt, Chd8, Dync1h1, Fkrp, Pik3ca, Ercc6 and Mecp2) encoding neurological disease-risk genes.

## Data Availability

The raw RNA-seq data (GSE141938) presented in this study can be downloaded from NCBI’s GEO database.

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
