# Peer review of "Unique and Specific m6A RNA Methylation in Mouse Embryonic and Postnatal Cerebral Cortices"

_genes, 2020, doi:10.3390/genes11101139_

Round 1
Reviewer 1 Report
# Comments to the authors:
In the manuscript entitled, “Unique and specific m6A RNA methylation in the mouse embryonic and postnatal cerebral cortex (genes-921493)”, the authors compared the distribution of m6A in the transcripts from tissues of embryonic and postnatal cortices. The study is more like a descriptive result, and should be further improved by providing more mechanistic investigation such as the information of m6A positions and their contribution to the regulation of (post)transcript.
# Major points:
1. There are about only one third of RNA with m6A in E stage and a quarter in P stage (Figure S2B and S2C). The major interest would be whether the absence of m6A is caused by or related to the absence of motifs or not. The issues should be finely dissected by answering the following questions:
1.1. Does the rest of RNA have the corresponding motif as identified in Figure 1B and 1C?
1.2. How general these two motifs can be identified from the transcripts (whole transcriptome) (compared to motifs with m6A)?
1.3. Do these two motifs have particular distribution along the gene structure (promoter, 5’-UTR, CDS..., compared to motifs with m6A)?
2. 40% of RNA has more than two sites of m6A (Figures S4C & S4D). How are they classified into the categories shown in Figure 2A if m6A sites fall in different regions? For example, in Figure 5, there are multiple peaks in CDS and 3’-UTR, how these would be treated?
3. Based on the text (line 144-146) and Figure S3A, it is difficult to interpret the molecular roles on m6A on the transcribed RNAs. There are multiple hypotheses stating the negative or positive roles of m6A (within different regions of mRNA) on the mRNA. To provide more information, the authors may consider moving the S3 to the main text, and sub-classify the mRNA according to the position of m6A, and evaluate how the correlation between the up-/down-regulation of mRNA and the positions of m6A.
4. The rationale behind the comparison of ‘proportions’ of peak distribution should be re-elaborated (Figure 2B). It is preferred to see the height of peaks for the same genes among different regions.
5. Related to the point 4, there is a significantly different ‘distribution’ in the regions of upstream 5~10K and 10~20K, how are they distinguished from intergenic regions (intgron)?? It will be assumed that the declared upstream 10~20K regions basically are intergenic regions. DO the distal regions (upstream or downstream 20K regions) overlap with other genes?
6. In Figure 4A and B, what is the RNA levels of Par3, Par6, Lats1, Tead1.. (all genes shown in Figure 4A and B)? Are these genes harboring only one m6A site or the motifs shown in Figure 1B and C?
7. Similar questions from point 6 should be applied to Figure 5.
8. Methods are not well described. For example, whether the ribosomal RNA was removed or not for the MeRIP, and/or how the FIRE was conducted.
## Minor points:
- Line 54, underscore should be removed for immuno’p’recipitation.
Author Response
Dear reviewer,
We are appreciated for your kindly recommend. The point-by-point response to your comments is presented in WORD file.
Please see the attachment. Thank you.
Sincerely,
Tao Sun

Reviewer 2 Report
This is a descriptive study documenting m6A methylation sites in RNAs from mouse embryonal brain (E12.3 – 13) and postnatal brain (P14). Methylation was measured by m6A-MeRIP (i.e. immunoprecipitation with a commercial antibody) followed by high-throughput sequencing. The paper presents the bioinformatic analysis of the data and reports to number of m6A sites at the two developmental stages, number of peaks per RNA, location within the RNA and functional categorization of RNAs that carry m6A.
My major concerns are 1) that the study contains no validation of whether the reported m6A sites truly represent m6A (rather than m6A(m) or antibody artefact), 2) that the first part of the study reports m6A that is not normalized to input and 3) that a comparison to prior studies of m6A sites in RNAs is missing. However, if these shortcomings can be addressed, this study can provide interesting information regarding the presence of m6A RNA methylation in tissues of the developing brain.
Individual points:
- The first part of the results section (up to line 175) describes the analysis of the m6A-seq data without the normalization to input RNA. This information should be omitted, since only peaks that are enriched relative to input are meaningful.
- The motif search in the m6A peaks was performed with the non-normalized peaks (lines 152 – 159). This analysis should be repeated, but with only the peaks that are enriched relative to input. Also, how do the identified motifs compare to previously identified motifs of m6A?
- The authors report the number of sequencing reads they obtained and analyzed (line 132). 1) Are these the reads for the triplicates taken together? I would like to know what the coverage for each sample was. This is important, because only if the coverage is at least 10 – 30x can meaningful statements be made about the number of peaks per transcript (c.f. PMID 32313079).
- For their m6A meRIP, the authors use an antibody from Synaptic Systems (Cat No. 202 003, a polyclonal antibody). The authors make no attempt to determine whether the peaks they identify are truly m6A peaks, or whether they are precipitation artefacts of the antibody. This could be done by depleting the methyltransferase (Mettl3/ 14) and determining whether the enrichment is reduced. Also, this particular antibody has previously been shown to detect m6A(m), not only m6A (PMID 26121403). Have the authors made any attempt to validate the specificity of the antibody? At least they should acknowledge that they cannot distinguish between m6A and m6A(m).
- The authors state that there are more m6A-methylated RNAs in the embryonic (E) than the postnatal (P) stage. The numbers, however, are very similar (Fig. 2A). How have the authors determined whether the difference is statistically significant? In Fig. 2B, I do not know what the bars are supposed to represent (why are these not just a single value?). The figure legend of Fig. 2B lacks the description of the individual abbreviations of the x-axis in the figure.
- The authors proceed to a GO analysis of the genes whose RNAs carry m6A in the E or P stage (Fig. 3). However, merely showing the number of genes in the individual categories is not meaningful. The authors should show the respective p-values in Fig. 3.
- In Fig. 4, 5 and 6, examples of m6A in RNAs are shown. Are the figures from a single replicate? How do the peaks look in the replicates? What is the scale of the x-axis?
- There are many studies in the scientific literature documenting m6A RNA methylation in RNAs of certain tissues, conditions etc. The present study lacks any comparison of their data to that of other studies. I think it would be important to know how big the overlap of m6A methylation to other tissues, conditions, studies etc is. This could give us a sense of how much variability there is. This variation could be biological or technical.
- Line 322 has the following statement: “Our analyses indicate that most disease-risk genes associated with neurological diseases are modified by m6A methylation with a variation of m6A binding sites.” How do the authors come to this conclusion? I cannot find the quantification in the ms. What is “most”? What fraction of those RNAs carry m6A?
Other:
While the language of the ms is mostly ok, there is one glaring mistake that is made several times and urgently needs careful revision. The authors find that many RNAs encoding transcription factors carry m6A. Across the ms, they then several times make the erroneous statement that the transcription factors are methylated, which of course is not true. Some examples:
Line 56: “and reveal specific methylation patterns of transcription factors…”
Line 273: “…next focused on analyzing m6A methylation status in cortical-specific transcription factors …”
Line 292: “…transcription factors with m6A sites…”
Line 366: “We have found that these transcription factors are highly methylated at the m6A site…”
The ms contains several instances of imprecise description and overstatements. Some examples:
Line 222: What is “genetic factor binding”? The term is not meaningful.
Line 307: “…we detected 4 pathogenic P-SMRs involved in Alzheimer’s disease…”
Line 21: What are “pathogenic genes”? Remove the word “pathogenic”.
This is an overstatement, because there is no evidence in this ms that the P-SMRs (i.e. m6A sites in postnatal RNA) are pathogenic. Remove the overstatement.
Line 348: “…Our study further confirms that m6A methylation plays crucial roles in regulating RNA splicing, stability and even translation efficiency.” Remove this overstatement, because this study shows no evidence of functional roles of m6A.
Line 370: “Our finding indicates that maintaining proper m6A methylation is required for precise regulation of gene expression and normal brain formation.” Remove this overstatement. This ms contains no functional characterization of m6A.
Author Response

(The authors gave the same response as above.)

Round 2
Reviewer 2 Report
Review Zhang et al, Revision, Sept. 2020
The authors have taken my suggestions into consideration and have improved and clarified the manuscript. The wording is still wrong in some instances. The authors appear not to have understood my criticism.
I suggest the following corrections:
1) “Our analyses indicate that the targeted genes associated with neurological diseases are modified by m6A methylation with a variation of m6A binding sites.” (Line 368).
Change to:
“Our analyses indicate that the targeted RNAs associated with neurological diseases are modified by m6A methylation with a variation of m6A binding sites.” (Line 368).
2) “…. reveal specific m6A patterns carried by transcription factors and disease-risk genes in the nervous system.” in line 56-57.
Change to:
“…. reveal specific m6A patterns carried byRNAs encoding transcription factors and disease-risk genes in the nervous system.” in line 56-57.
3) “we next focused on analyzing m6A-harboring status in cortical-specific transcription factors with m6A binding sites near stop codon and the 3’UTR.” in line 316-318.
Change to:
“we next focused on analyzing m6A-harboring status in RNAs encoding cortical-specific transcription factors with m6A binding sites near stop codon and the 3’UTR.” in line 316-318.
4) “…. were transcription factors harboring m6A sites in the 5’UTR region….” in line 336.
Change to:
“…. were RNAs encoding transcription factors harboring m6A sites in the 5’UTR region….” in line 336.
5) “We have found that these transcription factors harboring the m6A sites,…..” in line 415.
Change to:
“We have found that the RNAs encoding these transcription factors harboring the m6A sites,…..” in line 415.
Author Response

(The authors gave the same response as above.)
